# Influence of Systematic Gaze Patterns in Navigation and Search Tasks with Simulated Retinitis Pigmentosa

**DOI:** 10.3390/brainsci11020223

**Published:** 2021-02-12

**Authors:** Alexander Neugebauer, Katarina Stingl, Iliya Ivanov, Siegfried Wahl

**Affiliations:** 1ZEISS Vision Science Lab., Institute for Ophthalmic Research, Eberhard-Karls-University Tuebingen, 72076 Tuebingen, Germany; siegfried.wahl@uni-tuebingen.de; 2Center for Ophthalmology, University Eye Hospital, Eberhard Karls University Tuebingen, 72076 Tuebingen, Germany; katarina.stingl@med.uni-tuebingen.de; 3Center for Rare Eye Diseases, Eberhard Karls University Tuebingen, 72076 Tuebingen, Germany; 4Carl Zeiss Vision International GmbH, 73430 Aalen, Germany; iliya.ivanov@zeiss.com

**Keywords:** retinitis pigmentosa, visual performance test, visual field loss, vision impairment, goal-directed walking, visual search, virtual reality, gaze training

## Abstract

People living with a degenerative retinal disease such as retinitis pigmentosa are oftentimes faced with difficulties navigating in crowded places and avoiding obstacles due to their severely limited field of view. The study aimed to assess the potential of different patterns of eye movement (scanning patterns) to (i) increase the effective area of perception of participants with simulated retinitis pigmentosa scotoma and (ii) maintain or improve performance in visual tasks. Using a virtual reality headset with eye tracking, we simulated tunnel vision of 20° in diameter in visually healthy participants (*n* = 9). Employing this setup, we investigated how different scanning patterns influence the dynamic field of view—the average area over time covered by the field of view—of the participants in an obstacle avoidance task and in a search task. One of the two tested scanning patterns showed a significant improvement in both dynamic field of view (navigation 11%, search 7%) and collision avoidance (33%) when compared to trials without the suggested scanning pattern. However, participants took significantly longer (31%) to finish the navigation task when applying this scanning pattern. No significant improvements in search task performance were found when applying scanning patterns.

## 1. Introduction

Retinitis pigmentosa (RP) describes a subset of diseases that lead to severe concentric loss of vision (“tunnel vision”) in the peripheral field of view (FoV) [1,2,3]. It affects approximately 0.03% of the world population and is for the majority of cases not curable [4]. Although patients with RP oftentimes retain their normal visual acuity until late stages of the disease [5], the loss of peripheral vision was shown to severely limit the ability of patients to safely navigate and avoid obstacles [6,7]. However, the optic flow and judgement of direction is not affected by the decreased FoV [8,9], leading to the assumption that the decrease in navigation performance is caused mainly by a lack of obstacle awareness. This is further supported by the findings of F. Vargas-Martin and E. Peli [5], who showed that RP patients have a decreased average horizontal gaze amplitude compared to normal-sighted subjects. This seemingly contradictory behavior is assumed to originate from the lack of visual stimuli in the periphery, as gaze movement was shown to be guided by attention [10], and the target of a saccade rarely lies outside of the visual area. It must be noted, however, that other studies such as that of Turano et al. [11] have found contrasting results where the standard deviation of the gaze is significantly higher in RP patients compared to the visually healthy control group. Still, it can be assumed that in order to account for the lack of peripheral sight, larger gaze amplitudes are required in order to recognize obstacles and navigate safely. Only the area that was covered by gaze within a certain amount of time has the potential to give information to people with tunnel vision in a similar way that the peripheral FoV gives information to visually healthy individuals. This “gaze area over time” will in the following be called “dynamic field of view” (DFoV).

The only approved causal therapy for RP has been available since 2017 in the USA and since 2018 in Europe for retinitis pigmentosa caused by bi-allelic mutations in the gene RPE65 [12]. Although there are promising studies in the field of gene therapy for further genotypes in RP that could prohibit the progressive loss of the peripheral field [13,14,15], they are still in early research phases and do not guarantee the rehabilitation of the visual field. Most cases of RP are detected only after symptoms already appeared, at which point the damage of the degenerated peripheral photoreceptors is almost always irreversible [13,14]. Other studies have investigated the use of external, head-mounted displays to artificially increase the FoV [16,17,18,19,20,21,22] by compressing a larger visual area into the remaining FoV of the patient.

A third approach, which our study focuses on, are methods to guide the gaze through training of voluntary and controlled saccades [23]. These gaze movements are typically slower than “natural” reflex-like saccades, so the question remains whether it can be applied as a neural plasticity training to become as fast and nonintrusive as natural saccades, which is suggested by the effectiveness of neural plasticity training on saccadic adaptation in visually healthy persons [24,25,26,27]. In a study carried out by Ivanov et al. [23] it was shown that training with a visual search task on a computer display, designed to demand and encourage larger saccades, could partially lead to navigation improvement in RP patients as they walked longer durations at their preferred walking speed after the course of six weeks of training. These findings are in line with similar studies investigating compensatory training for patients with hemianopia—the loss of the left or right hemisphere of the FoV, usually caused by a stroke—in which gaze training was also found to improve detection and reaction rates in different visual tasks [28,29,30,31]. However, the number of collisions with obstacles during the navigation trials did not decrease in Ivanov’s study. This shows that the training of saccades has indeed influence on the navigation, but larger saccades alone might not suffice to improve the perception of obstacles, which suggests the addition of a more systematic gaze pattern to the training.

Being able to perceive a larger visible area may not only be important for obstacle awareness but for larger-scale orientation as well. Landmarks are an essential aid to remember a route or navigate unknown routes based on descriptions or maps. The selection of such landmarks is closely connected to the gaze behavior [32], and so an increased DFoV is likely to increase the number of detected landmarks and thus improve orientation and route learning.

When trying to improve the DFoV of patients with tunnel vision through systematic scanning patterns, the question arises as to which type of pattern is most beneficial. It can be assumed that the natural gaze behavior of visually healthy people is already optimal, as it is guided by the stimuli from peripheral vision. This, however, may not be the case for patients with tunnel vision or other conditions that occlude the FoV, as is suggested by previous studies of gaze behavior of people with visual impairments [23,28,30,33]. Here, no stimuli from the periphery of the FoV exist to draw the gaze toward important focus points in the scene. In these cases, the gaze direction is much more crucial for the general awareness of an object or point of interest in the scene.

In our study, we therefore investigated whether systematic scanning patterns have an influence on the DFoV as well as the performance in different visual tasks in participants with simulated tunnel vision and if so, which systematic scanning pattern has most potential to lead to improvements in visual tasks when being applied in training. 

## 2. Materials and Methods

### 2.1. Ethics

This study was proposed to and approved by the ethics committee of the Institutional Review Board of the Medical Faculty of the University of Tübingen (628/2018802) in accordance with the 2013 Helsinki Declaration. All participants signed informed consent forms. 

### 2.2. Software and Hardware Specifications

The experiment was performed in a virtual reality (VR) environment. By this, larger viewing angles than with a standard screen can be achieved. Furthermore, head rotations can be detected within the simulation to measure the influence of both head and eye movements in the different scanning patterns. The virtual experimental environment and visual tasks as well as the tunnel vision simulation were created for this study using the game engine Unity3D, Version 2019.2. 

The VR headset on which the virtual content was displayed was the FOVE 0. It had a 70 Hz screen refresh rate and a 120 Hz eye-tracking refresh rate, eye-tracking accuracy of 1°, and a visual field of 100°, according to its technical data sheet [34]. Manual testing found that the visual field per eye measured 81–85° in horizontal direction and 88–91° in vertical direction. The latency of the eye-tracking of the headset was estimated to be between 20 and 50 ms [35]. The frame rate of the application itself did temporarily drop to ~40 frames per s depending on the number of obstacles on screen and the amount of rapid head movement. The wire connecting the headset to the laptop had a length of 3 m. A wireless Microsoft Xbox One X controller was used as an input device for the visual task execution.

### 2.3. Scanning Patterns

Two artificial gaze patterns were evaluated by comparing both their DFoV and visual task performance to each other, as well as to a free gaze condition where participants were asked to use their gaze as they normally would. Both scanning patterns were designed with focus on efficient use of the remaining FoV. The first one was the “left–right pattern”, where most saccades are of horizontal nature (Figure 1a). This maximized the covered area, as no point of the FoV was covered twice within one pattern. The distance between each gaze line should ideally have been equal to the angular field of view of the participant. This pattern was suggested by Ivanov et al. [23] and was also used in visual training studies for patients with hemianopia [28].

The second gaze pattern (Figure 1b) was a radial pattern in which saccades always occurred between the center view and the periphery. Here, the per-time area coverage was not as optimized as in the left–right pattern. However, it allowed participants to keep track of the area in the center of the FoV, which may have been beneficial especially in navigation scenarios in order to follow the desired walking direction. It also allowed participants to shift gaze more freely into different directions of the periphery. 

The different scanning patterns were shown as lines to participants at the start of the respective session. After showing the scanning pattern, the participants were asked to repeat the pattern, first with guidelines and then again without. This was repeated multiple times until the participants felt confident in being able to repeat the pattern consistently without guidelines. It was explained to the participants that they were not required to continuously follow the scanning patterns at all times and were allowed to fixate points of interest when necessary, but they were instructed to substitute any large gaze movements with the previously studied scanning pattern. The consistent execution of scanning patterns was supervised by the experimenter, who could follow the gaze of the participants on a laptop display. When necessary, participants were reminded between trials to follow the scanning pattern. 

### 2.4. Study Population

The study population consisted of nine participants (four male, five female) between 19 and 27 years of age (average 23.2 ± 2.49 years) for the main group as well as three participants (all female) in the control group (aged 23 to 27 years, average 24.7 ± 2.08 years). No additional control participants could be recruited due to restrictions for participant studies introduced in response to the SARS-CoV-2 pandemic. All participants were self-reportedly visually healthy, with two participants stating to occasionally wear glasses in lectures or while driving. Glasses could not be worn during the experiment as they did not fit under the VR headset; however, it was made sure before the experiment that the search task targets could be recognized effortlessly by the participants. Table 1 shows the participants’ age, sex, and vision correction as well as their experience with controllers and VR headsets.

### 2.5. Mesasured Parameters

To evaluate the effectiveness of the scanning patterns, both DFoV as well as visual performance were assessed. The DFoV was measured as the average area covered by the restricted FoV of 20° over a moving window of 3 s, measured once per s. We roughly estimated based on the findings of Peli et al. [36] that at normal navigation speed between 1 and 2 m∙s^−1^, 3 s was sufficient to react to most obstacles, including pedestrians. However, the number was mostly arbitrary as it was only used to compare the DFoV of different scanning conditions against each other. Results for the DFoV over 1, 5, and 10 s durations can be found in the Appendix A. The visual performance was measured by the time required to complete a navigation trial and the number of obstacle collisions within one navigation trial, or as the correctness of the number of targets found in the search task.

We also observed the variability of gaze direction for both eye-only movement and the total gaze, including head rotation. These parameters described the standard deviation of the horizontal and vertical position of the gaze and allowed a more detailed analysis of how different scanning patterns as well as different visual tasks influenced the gaze behavior.

At the end of each session, a study questionnaire was filled out in which the participants were asked to rate their physical and mental strain, their subjectively perceived success in the visual tasks, as well as their expectations for long-term training with the respective scanning pattern. 

### 2.6. Experimental Setup

The study was carried out over the course of three sessions between 60 and 90 min. In each session, one of three different conditions were selected, either the left–right eye scanning pattern, the radial eye scanning pattern, or no systematic scanning pattern. Those three conditions were pseudo-randomized such that all three conditions were equally distributed between the three sessions, thus minimizing the influence of task learning effects. 

In each session, the participants were asked to do a total of 20 navigation trials and 40 search trials, split into two sets of 10 navigation and 20 search trials, respectively, and a 15 min break in between. Additionally, in the first session and before the start of the trials, the participants were introduced to the headset and were able to familiarize themselves with the headset and the controls of the navigation task. In both sessions where a scanning pattern was applied, the respective pattern was shown to the participants and trained for 5 to 10 min before the trials began. The participants remained seated during the experiment. The movement in the navigation task was done using the controller; however, participants were required to turn in their swivel chair in order to change the direction they were facing, thus increasing the immersion of navigation. The control group performed the same visual tasks with the same scanning patterns as the main group, with the only difference being that they carried out the tasks with unrestricted FoV. 

### 2.7. Visual Tasks

The navigation task was done by applying a virtual, randomized environment resembling a corridor of 8 m width with parquet floor and white walls, as is seen on the left side of Figure 2a. Each environment consisted of eight 8 × 8 m tiles in randomized order, out of which two were the starting and end tile, two resembled straight corridors, two had left corners and two had right corners, allowing for a total of 120 different corridor layouts (examples being shown in Figure 3). In addition, at the beginning of each new tile, a random obstacle out of a selection of 15 different obstacles was instantiated, such as different walls, low-hanging bars, or simulated pedestrians. Lastly, some low-height obstacles such as barrels or small fences were instantiated at different positions on the tiles. This variety of obstacles encouraged gaze variation both horizontally and vertically and even required participants to adapt to sudden changes of the environment. 

The randomized nature of both the layout of the corridor and the obstacles allowed for a near infinite number of environments, precluding the possibility that obstacle courses were repeated and recognized in later trials. Participants were able to virtually walk at a maximum speed of 3 m∙s^−1^ or ~10 km∙h^−1^ with an acceleration of 3 m∙s^−2^. However, participants could freely adjust their walking speed below that limit by not pushing the thumb stick of the controller to full extent. Collisions with walls or obstacles were indicated by a bouncing sound effect and the participant’s avatar was knocked back by up to 1 m depending on collision speed, thus preventing movement through obstacles and giving participants space to adjust their course without the risk of repeatedly colliding with the same obstacle. The simulated pedestrians moved in intervals between 4 and 10 s at a speed of 2 m∙s^−1^, allowing participants to avoid collision as long as they were aware of their presence and movement.

The search task trials started by presenting a random digit to the participant with the additional information to search for this number in the upcoming screen. Then, 30 random digits were instantiated, distributed in a spherical field of 95° × 60° with a 3 m radius (see Figure 2b). The participants were given 20 s to search the scene and were instructed to press a button on the controller whenever they spotted a target with the specified target digit. After 20 s, all digits disappeared, and the next trial started. A trial was marked as correct if after 20 s the number of button presses matched the number of target digits.

### 2.8. Questionnaire

The questionnaire was structured following the basic layout of a system usability scale by John Brooke [37], however with four instead of five choices (very little, little, high, very high) in order to prevent indecisive participants from choosing the middle option. The questions were adjusted to fit the research focus.

### 2.9. Statistical Methods

A total of 545 navigation task trials and 1070 search task trials, split among the nine participants and three different scanning pattern conditions, were evaluated. In the control group, the results of 180 navigation task trials and 358 search task trials were acquired and evaluated. For the analysis of the effects of systematic scanning patterns on DFoV, gaze variability, as well as navigation duration, linear mixed-effect models were applied. This allowed the analysis to account for the varying intrinsic performances of participants as a random factor. The models considered both random intercept and random slope of the random factor, since consistent data across all participants could not be assumed. Additionally, the models considered the trial number as a fixed effect. For the analysis of collisions in the navigation task, a zero-inflated Poisson regression was applied. Both trial number and trial time were considered as fixed effects. The rationale for this will be explained in the Discussion. To analyze the search task performance, a binomial generalized linear mixed-effects model was used, where participants were again considered as a random factor. A detailed summary of the statistical results for the applied models is found in Appendix C, Table A1, Table A2, Table A3, Table A4 and Table A5. Due to the pseudo-randomized order of scanning pattern conditions, which ensured an equal distribution of the conditions over the three sessions, the session in which a trial took place was not included as a factor in the analysis. All averages are given including the standard deviation. Error bars also show standard deviations. The analysis was done in R using the lme4 and pscl package. 

The questionnaire is presented as averages. A statistical test for significance was not feasible due to the small sample size of only one answer per participant and scanning pattern.

## 3. Results

### 3.1. Navigation Results

Figure 4 shows the results for DFoV, number of collisions, and time required to finish the task, averaged over all navigation trials of the main participant group and control group, respectively.

All values of DFoV are given as percentage of a 200° × 150° field. This field was used as reference as it roughly describes the horizontal and vertical angle of the FoV of a visually healthy person. The average DFoV of all trials of the main participant group was 4.59 ± 0.73%. Without any suggested scanning pattern, the DFoV was 4.39 ± 0.73% (control group 4.50 ± 0.70%). With left–right-pattern being applied, the DFoV was 4.82 ± 0.74% (control group 6.71 ± 1.85%), and, with the radial pattern applied, it was 4.57 ± 0.73% (control group 6.77 ± 1.53%). If only trials without any collisions are considered for a collision-independent comparison, the DFoV was 4.31 ± 0.73% for 115 free trials, 4.86 ± 0.73% for 122 left–right trials, and 4.54 ± 0.73% for 112 radial trials. In both cases, the left–right pattern increased the DFoV by approximately 10% compared to the no-pattern trials (*p* = 0.034 for all trials; *p* = 0.02 for no-collision trials), and the radial pattern increased the DFoV by approximately 5% (*p* = 0.269 for all trials; *p* = 0.096 for no-collision trials). The average number of collisions per trial without applied scanning pattern was 0.73 ± 0.85 (control group 0.05 ± 0.22). It was significantly lower in both scanning pattern conditions at 0.49 ± 0.70 (*p* < 0.001) (control group 0.1 ± 0.32) in the left–right pattern trials and 0.58 ± 0.75 (*p* < 0.001) (control group 0.12 ± 0.35) in the radial pattern. The average time required for a navigation trial was found to be significantly lower when no scanning pattern was applied (37.3 ± 12.2 s, *p* = 0.0011 for left–right; *p* = 0.0017 for radial; control group 24.4 ± 5.08 s), while trials with applied left–right pattern had an average trial duration of 48.8 ± 15.8 s (control group 28.1 ± 6.70 s) and trials with radial pattern 48.5 ± 16.8 s (control group 28.7 ± 7.95 s). 

The number of collisions was decreased for trials with left–right scanning pattern even when accounting for the differences in average trial durations. The zero-inflated Poisson regression of collisions showed an estimate for the count model coefficients of the left–right condition of −1.09 with a standard error of 0.16 and for the radial condition an estimate of −1.10 with a standard error of 0.14. In both conditions, the *p*-values were below 0.001. The implications will be explained in more detail in the Discussion. The lines in Figure 5 show the average number of collisions in each scanning condition for trials of similar duration in a moving average. The moving average was applied only to the plot in Figure 5 to provide more clarity and had no influence on the statistical analysis. A breakdown of DFoV, average collisions, and trial duration for individual participants is found in Appendix B, Figure A1. Summaries of the analysis results for the navigation trials are found in Appendix C, Table A1, Table A2 and Table A3. Raw data for DFoV, number of collisions and trial duration in navigation trials is found in the Appendix A.

The variability of gaze direction describes the standard deviation of horizontal and vertical gaze angles. Figure 6 displays the variability of the total gaze that describes the actual gaze direction of eye rotation and head rotation combined, compared to body rotation, as well as the variability for only eye movement and head rotation independently. Since the experimental setup only allowed tracking of head rotation and eye position but not full body rotation, the body rotation was estimated as the 3 s average of head rotation, assuming that a person facing in a direction for more than 3 s while walking would turn their body accordingly. Both total gaze variability and head rotation variability were measured in reference to the estimated body rotation.

Horizontal gaze variability was almost 3 times as high as vertical gaze variability in all three scanning pattern conditions, with an average of 18.26 ± 3.44° (control 24.08 ± 4.24°) horizontal gaze variability and 6.37 ± 1.90° (control 7.13 ± 2.66°) vertical gaze variability. Between different gaze pattern conditions, only small differences in both horizontal and vertical gaze variability were found in the main participant group, with the most relevant differences being displayed in the decrease of horizontal head rotation variability in the left–right (*p* = 0.072) and radial (*p* < 0.001) pattern. In the control group, there was a strong increase in vertical gaze variability, with both total gaze and eye position variability increasing by 73.2% to 81.5% in both scanning patterns compared to no-pattern trials. This increase was not found in the main participant group. Raw data for navigation trials can be found in the Appendix A.

### 3.2. Search Task Results

In the search task, the average DFoV over 3 s was higher in left–right trials (*p* = 0.099) with an average of 3.53 ± 0.75% (control 3.85 ± 1.03%) compared to the trials with no applied scanning pattern with 3.29 ± 0.69% (control 3.37 ± 1.04%) DFoV. The average DFoV of the radial scanning pattern trials was lower (*p* = 0.327) at 3.16 ± 0.66% (control 2.95 ± 0.76%), as can be seen in Figure 7. This trend was found in both main and control group. No significant difference between the left–right scanning pattern trials and the no-pattern-trials could be found in the search performance—the ratio at which the number of targets in the search task was correctly identified—with 75.9% correct trials in the no-pattern-condition and 76.5% in the left–right pattern condition (*p* = 0.804). The radial condition had a lower ratio of correct trials compared to the no-pattern condition (*p* = 0.089) with only 69.5% correct trials. There was, however, a significant effect between DFoV and search performance (*p* = 0.0016). The results for individual participants are found in the Appendix B, Figure A2. Summaries of the analysis results for search trials are found in Appendix C, Table A4 and Table A5. Raw data of DFoV and performance in search trials is found in the Appendix A.

The gaze variability in the search task that is displayed in Figure 8 shows larger differences between scanning pattern conditions, but less difference between main and control group compared to the gaze variability during navigation task. Most notable is the difference in eye position and head rotation variability between the radial pattern trials and the two other scanning pattern conditions. The main group shows an increase in horizontal eye position variability in radial patterns of 47.5% compared to the no-pattern trials, but a 57.5% decrease in head rotation variability, with smaller, but similar differences found in vertical direction. This contrasts with the left–right pattern, which is more in line with the no-pattern variability of eye position and head rotation. The raw data for gaze variability in all search trials is found in the Appendix A.

### 3.3. Questionnaire Results

Figure 9 displays the rating reported by the participants on their perceived performance in the two visual tasks, as well as the rating of physical and mental strain, meaning the strain on the eyes as well as required concentration. 

Additionally, the question “Which of the two eye scanning patterns is more intuitive?” revealed that all nine participants as well as the three participants of the control group preferred the left–right pattern. To the question “Do you believe that it is possible to adapt to at least one of the scanning patterns in a way that it feels natural and unobtrusive to use?”, three participants answered that they could imagine it for both scanning patterns, five participants could only imagine it for the left–right scanning pattern, and one participant could not imagine either of the scanning patterns being perceived as natural and unobtrusive. Two of the participants mentioned that even during the study they felt the effects of adaptation. Individual results for the participants can be found in the Appendix A.

## 4. Discussion

We investigated whether it was possible to increase the dynamic field of view of participants with simulated tunnel vision by applying systematic eye scanning patterns and evaluated in which way these eye scanning patterns influenced the performance in two visual tasks—a navigation task and a search task. For that, a software for virtual reality was developed that utilized the built-in eye tracking functionality of the VR headset to simulate a FoV limited to 20° in participants with healthy eyesight. 

It was found that applying a systematic “left–right” eye scanning pattern did increase the DFoV of the participants by approximately 10% compared to the DFoV without scanning pattern. Further, the number of collisions with obstacles in a virtual navigation task was significantly reduced with applied scanning patterns, with the left–right pattern reducing them by 32.9% and the radial pattern by 20.5%. The time required to navigate through the obstacle parkour however increased by 30.8% (left–right) and 30.0% (radial) when applying the scanning patterns. Partially, this can be explained by a shift in the speed–error tradeoff often found in psychophysical experiments. The confrontation with a new task—applying the systematic gaze patterns—and the resulting reduction in confidence compared to the no-pattern trials likely led to a shift toward error prevention at the cost of speed [38]. However, by analyzing the number of collisions in a model that considered the trial duration as a fixed effect, we found that the scanning patterns had an influence on the collision reduction that could not be attributed only to the difference in average trial duration and thus the speed–error tradeoff. Figure 5 further supports this finding, as it shows that in trials of similar duration, the average number of collisions was typically lower in scanning pattern trials—especially the left–right pattern trials—compared to the no-pattern trials.

The performance in the visual search task, measured by the rate of trials in which the correct number of targets was identified by the participants, did not improve with applied scanning patterns and was even decreased for the radial scanning pattern by 9.3% compared to the trials with no scanning pattern. The DFoV in the search task was increased by 7.3% for the left–right pattern but decreased by 4.0% for the radial pattern compared to trials without scanning pattern. The analyses of these effects did not show significance, but the numerical trends suggest that out of the two scanning patterns tested, the left–right pattern may have led to similar or better results than the radial scanning pattern in all aspects. It was also favored by participants in terms of intuitiveness, perceived success, mental and physical strain, as well as expected potential to naturally adapt to this scanning pattern. It can be assumed that the left–right scanning pattern fell more in line with natural eye movements due to the horizontal sweeping motion of the gaze known from other visual tasks, such as reading. Further, the left–right pattern showed improvements or similar results in all aspects except trial duration compared to the no-pattern condition, indicating that scanning patterns were indeed able to increase the performance in visual tasks to a degree.

The results show that the DFoV was on average around 30% higher during the navigation task than during the search task. This can be explained by the fact that it is more difficult to recognize a certain number and distinguish them from other numbers rather than just noticing the existence and dimensions of an obstacle [39]. Furthermore, it is likely that increasing the average distances between the targets of the search task by either reducing the number of targets or increasing the field size would increase the DFoV.

The total gaze variability shows that the increase in DFoV in the left–right scanning pattern seems to have different causes based on the visual task. In the navigation task, the horizontal gaze variability did not increase significantly, whereas the vertical gaze variability showed an increase equal to that of the DFoV. Meanwhile, in the search task, the opposite was found in that only the horizontal gaze variability increased when applying the left–right scanning pattern. It was also found that scanning patterns—especially the radial scanning pattern—decreased the variability of head rotation and in return led to an increase in eye position variability. Kerkhoff et al. [40] found that an increase in head movements does not improve visual performance in patients with visual field disorders and can even reduce the effects of training. This suggests that the scanning patterns could improve the success of visual gaze training even further.

To the best of our knowledge, no other research regarding the influence of suggested eye scanning patterns on the visual performance in navigation and search tasks of participants with tunnel vision has been published to this point, apart from the preceding study by Ivanov et al. [23] that motivated this study. It is interesting to note that Ivanov et al. found a significant improvement of preferred walking speed of the participants after their training—which would be comparable to the navigation trial duration measured in this study—whereas no significant improvements in the number of collisions per trial were found. This contrasts with our findings, where the navigation trial duration increased when applying systematic scanning patterns, but the number of collisions decreased. This can be explained by changes in the speed–error tradeoff of the participants. It can be assumed that in the study by Ivanov et al., participants adapted to the new gaze behavior introduced to them over the course of the six weeks of daily training. Thus, they became more confident in the simultaneous execution of visual tasks and trained gaze behavior. Contrary to that, our study did not apply any training. Measurements were taken within an hour after the participants were first introduced to the new gaze behavior. This means that participants had very little time to adapt to the new gaze behavior and thus their confidence level likely was lower in the scanning pattern trials compared to the reference (the no-pattern trials). It was found that confidence has a direct influence on the speed-error tradeoff in subjects [38], where higher confidence levels result in a shift toward faster execution times and lower confidence levels result in a shift toward lower error rate. This is in line with the comparison of the results between the study by Ivanov et al. and ours. It further suggests that by applying saccadic search training in combination with systematic scanning patterns, it may be possible to improve both average walking speed and obstacle avoidance in patients with tunnel vision.

Comparison to studies that evaluated visual aiding devices, such as that of Luo et al. [19], Hicks et al. [20] or Angelopoulos et al. [22], is not feasible, since the goal of our study was not to find the full extent of visual performance improvement that was achievable with scanning patterns, which would require a much more extensive training, but only to compare different scanning patterns to each other and the performance without scanning pattern. It is thus unsurprising that our study did not find performance improvements of a similar magnitude to studies with visual aiding devices, such as a reduction of errors of 50% in both search task and obstacle avoidance [22] or an over 50% reduction in navigation trial duration over 10 trials [20] through augmented reality depth mapping. 

It has to be noted that the results are not fully representative of patients living with RP, as all study participants were visually healthy, and the limitation of the visual field was only simulated within the VR environment. Although the simulation of visual impairments in visually healthy participants was shown to induce similar behavior as their real counterparts [41], there are multiple aspects due to which results could vary if the experiment were to be repeated with real patients. First is the difference in experience with the condition. The participants of this study were all new to the experience of a limited FoV, and thus they did not have time to develop their own gaze strategies. Second, the use of a VR headset for the simulation brought its own difficulties. The FoV of the VR headset was limited to 100° independent of the limits of the simulated RP scotoma, which means that any viewing angle further than 50° to any side was not possible with eye movement alone. Lastly, the participants had to remain seated during both visual tasks, which especially in the navigation task resulted in further deviation from real-life scenarios. However, eye-tracking is a crucial aspect for both simulating tunnel vision as well as measuring the DFoV, and to our knowledge, wireless VR headsets with reliable eye tracking solution were not publicly available during the time of the experiments. It was thus necessary to use a wired solution combining controller navigation and body rotation. The choice to simulate RP in this way rather than to recruit real patients was made in view of a larger-scale follow-up study in which the effects of long-term scanning pattern training are assessed. We avoided recruiting patients at this point that would then no longer be able to participate in this larger-scale study without having an advantage compared to other participants due to the previous scanning pattern training.

## 5. Conclusions

Based on a setup simulating tunnel vision in a virtual-reality environment, we showed that the application of systematic gaze patterns can improve obstacle avoidance and dynamic range of the field of view. It was found that the scanning pattern focusing on straight, horizontal, sweeping eye movements (left–right pattern) led to an overall similar or better performance than the pattern based on radial eye movements from the center of the visual field to its periphery and back—and was also better accepted by participants. Based on these findings, the left–right pattern will be applied in a follow-up study. In this follow-up study, a six-week saccadic gaze training for patients with retinitis pigmentosa will be performed to investigate the effects of scanning pattern training on real-world navigation and obstacle avoidance.

## Figures and Tables

**Figure 1 brainsci-11-00223-f001:**
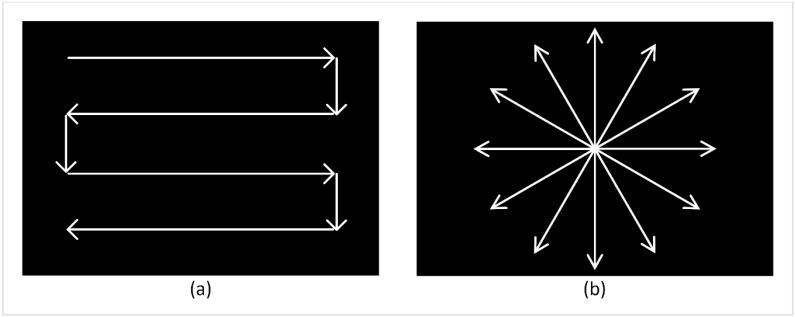
Visual representation of the left–right scanning pattern (**a**) and the radial scanning pattern (**b**) that participants were asked to follow during the visual tasks.

**Figure 2 brainsci-11-00223-f002:**
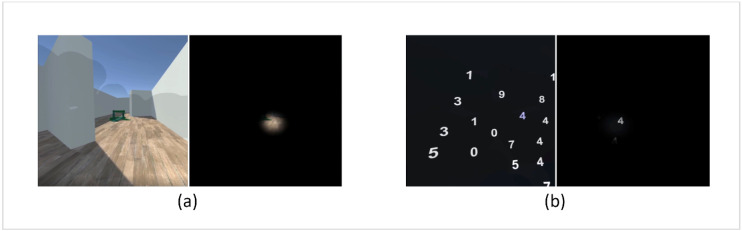
(**a**) Example of a navigation trial from participant’s view, without (**left**) and with (**right**) simulated tunnel vision. The slightly darkened area on the left side image indicates the area already covered by gaze for test and presentation purposes but was completely transparent during trials. (**b**) Example of a search trial from participant’s view without (**left**) and with (**right**) simulated tunnel vision. The digits are randomly distributed search targets. Both images (**a**,**b**) are captured from the scene render on the laptop screen. The content viewed by the headset user may have varied in proportions due to the influence of the headset’s lenses and different screen shapes.

**Figure 3 brainsci-11-00223-f003:**
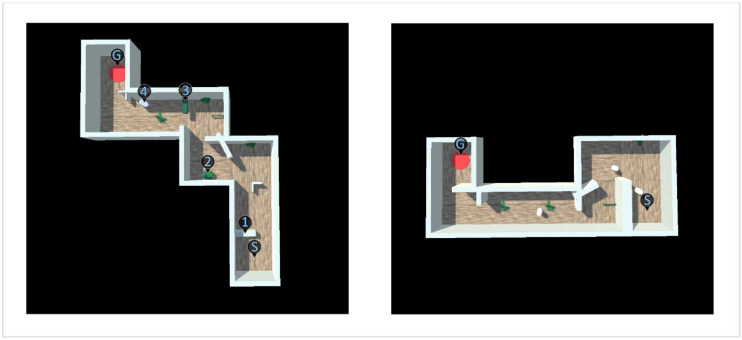
Top-down view of two examples of the randomized obstacle parkour used in the navigation task. The letter S indicates the starting point, the red circle marked with a G indicates the goal of the parkour. The blue digits mark examples of the four different obstacle types: 1 = large static obstacle/wall; 2 = low-height obstacle; 3 = obstacle hanging from ceiling; 4 = pedestrian.

**Figure 4 brainsci-11-00223-f004:**
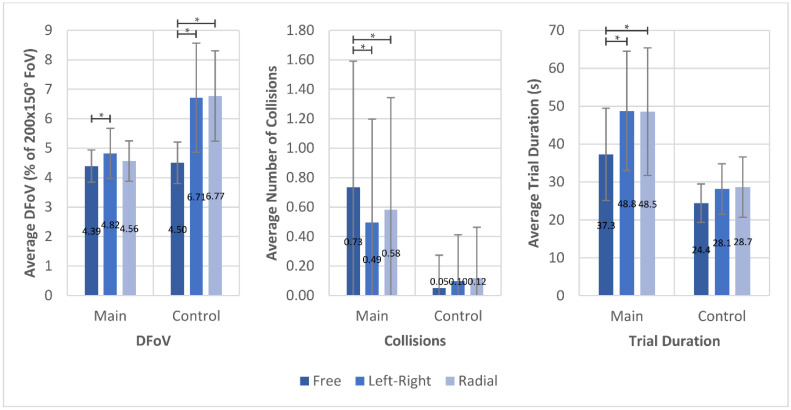
Comparison between average dynamic field of view (DFoV), number of collisions, and trial duration in the three different scanning pattern conditions during navigation for both main group and control group (* *p* < 0.05).

**Figure 5 brainsci-11-00223-f005:**
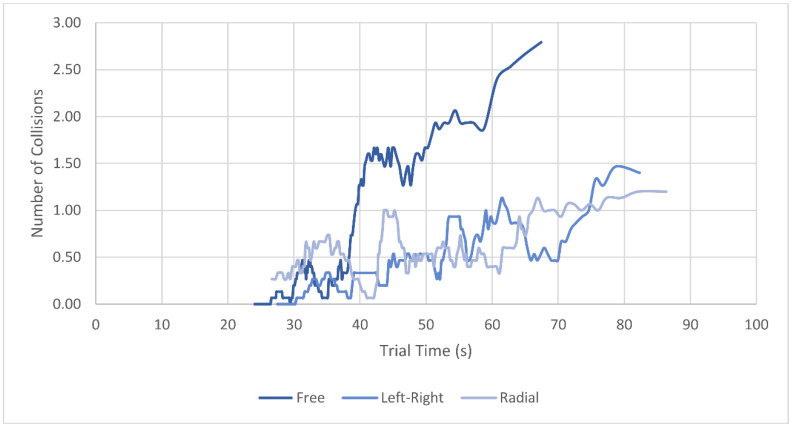
The moving average of the number of collisions by trial duration in the three conditions (period of 15). This visualizes how the number of collisions in trials of similar duration varied between the different scanning pattern conditions. The results shown are from the main participant group.

**Figure 6 brainsci-11-00223-f006:**
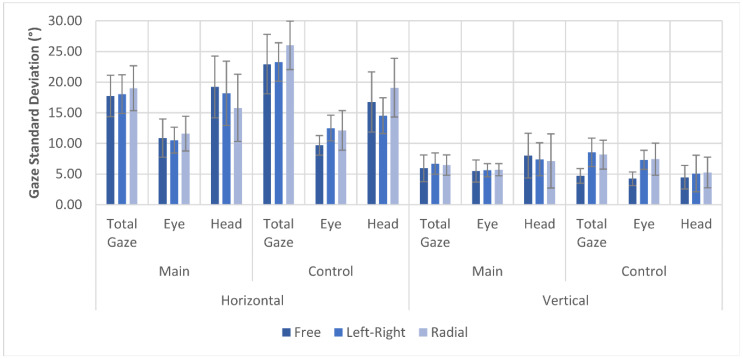
The average standard deviation of total gaze, eye position, and head rotation in the navigation task. Total gaze and head rotation were measured in reference to the estimated facing direction of the body.

**Figure 7 brainsci-11-00223-f007:**
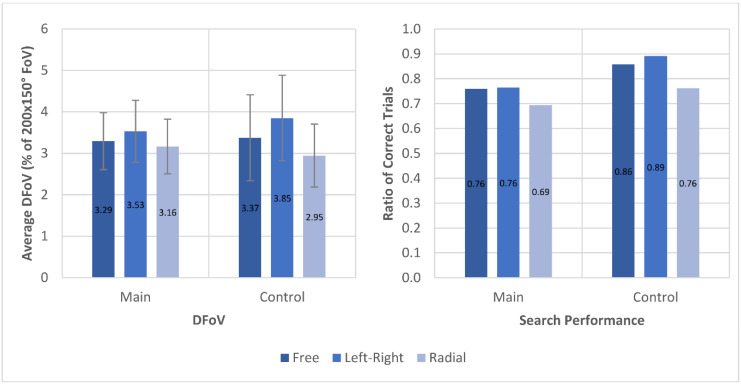
Comparison of average DFoV and ratio of trials in which the correct number of targets were found between the three different scanning pattern conditions during search task for both main group and control group.

**Figure 8 brainsci-11-00223-f008:**
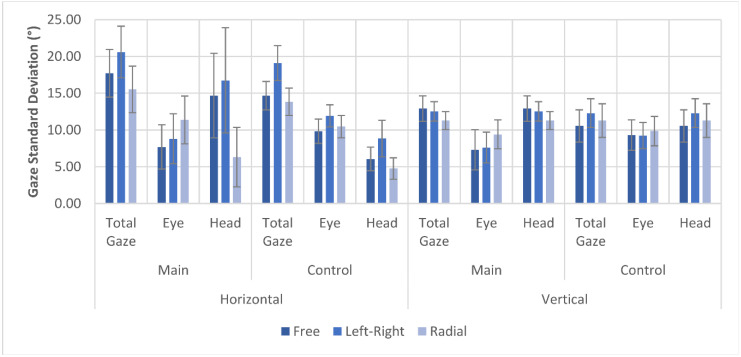
The average standard deviation of total gaze, eye position, and head rotation in the search task. Total gaze and head rotation were measured in reference to the estimated facing direction of the body.

**Figure 9 brainsci-11-00223-f009:**
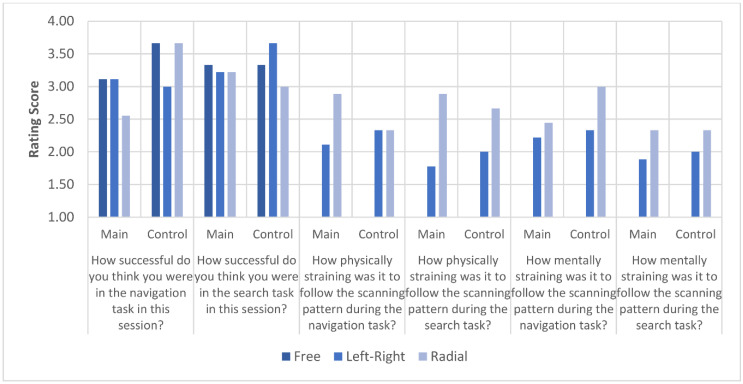
The results of the participant questionnaire, ranked from 1 = very low to 4 = very high.

**Table 1 brainsci-11-00223-t001:** Participant age, sex, vision correction as well as their previous experience with VR headsets and game controllers. A capital C marks participants of the control group.

Participant	Age	Sex	Vision Correction	VR Experience	Controller Experience
1	23	m	-	no	yes
2	25	m	-	no	yes
3	21	f	-	some	yes
4	24	f	-	no	no
5	27	f	-	no	some
6	20	m	G ^1^	no	some
7	23	f	-	some	some
8	19	f	-	some	some
9	23	m	G ^1^	some	yes
C1	27	f	-	some	some
C2	23	f	-	some	no
C3	24	f	-	no	some

^1^ occasionally wearing glasses.

## Data Availability

The data presented in this study are available in the Appendix A.

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
