# Peer review of "Influence of Systematic Gaze Patterns in Navigation and Search Tasks with Simulated Retinitis Pigmentosa"

_brainsci, 2021, doi:10.3390/brainsci11020223_

Round 1

Reviewer 1 Report

In “Influence of Systematic Gaze Patterns in Navigation and Search Tasks with Simulated Retinitis Pigmentosa”, Neugebauer and colleagues analyze the effect of simulated restriction of the visual field and gaze-pattern instructions on navigation in a virtual environment and visual search performance. For the main comparison of interest, the authors instructed participants to either move their gaze in a left-right sweeping pattern or in a radial pattern and conclude that the former leads to overall better performance. I am not a clinical researcher, so I could not comment on the relationship with the Retinitis Pigmentosa diseases that the employed conditions emulate. I think this study is a valid first step to answer the authors’ research questions, but I do have concerns about the data analysis and the amount of support for the conclusions. Overall, I would argue that before engaging in a six-week follow-up study with patients, stronger evidence is needed for the effectiveness of the manipulations.

Major

  1. Do the participants even follow the instructions? Were the gaze patterns required from them actually used consistently throughout the study? This is critical for the interpretation of the results, yet this is not reported.
  2. The authors conclude that the “left-right” pattern led to better performance. However, none of the tests in the navigation results shows this. That is, performance sometimes is numerically better, but none of the tests shows significant improvement.
  3. The authors apply a random intercept-only mixed-effects model. Such models excluding random slopes are known to inflate significance (Barr et al., 2013), although there could be reasons to use them if more complex models could not be supported by the data (Bates et al., 2015). This random intercept approach seems problematic here as individual participants data does not seem consistent in some of the measures.
  4. For collisions analyses, using a standard LMER model is a problematic move as the data is clearly non-normally distributed. Instead, a zero-inflated Poisson GLMER could be a good option for this kind of counting data.
  5. There is a lack of details on analyses shown in Fig. 5. Were the trials with all durations shown in the figure present in all participants and all conditions (i.e., were, for example, trials with >60 s duration present for all three conditions in all subjects?) or did binning made data imbalanced? How this was handled? What kind of statistical analyses were done on the data (only p-value is reported)?

Minor

  1. The screenshots in Fig 2 seem to indicate that the restricted FoV was much lower than 20°. If the full field is 100° (line 112), then 20° should be 1/5th of it. From the screenshots, it looks more like 2°.
  2. More details on the search task are needed. How did participants indicate, where is the target digit? Why the accuracy is so poor even in the control condition? For 20 s trial, I expect participants to be able to find the target nearly always.
  3. What is the rationale for analyzing gaze variability? It is not introduced in any way and only briefly discussed. Furthermore, wouldn’t different conditions change variability just by virtue of requiring different eye patterns?
  4. It’s not fully clear, how DFoV was measured. On line 158, the authors state “The DFoV is measured as the average area covered by the restricted FoV of 20° over the last three seconds”. Was it the last three seconds of a trial or the average with a 3 s. moving window across the trial? If only the last 3 s were used, what is the justification for discarding the rest of the data?
  5. The sentence on lines 370-373 is very hard to understand; please clarify it.

Reviewer 2 Report

My apologies for the slight delay in my review.

Generally, I think this is a well conducted and well presented study. I would have a few probably minor comments and suggestions for improvement.

  • Please explain why you chose to only test three control participants while testing nine in the main condition.
  • For the analysis in Figure 5, survival analysis may be more appropriate, in which you treat trials without a collision as censored data, and create 'survival plots' that plot the time until the first, second, third collision. This may answer a slightly different question than the one you address in figure 5, so I am open to arguments why your analysis is more appropriate.
  • Can you explain the differences in results between your study and the study by Ivanov et al? 
  • Please explain how you checked to which extent participants succeeded in reproducing the instructed eye movement patterns.
  • In the description of the mixed effects analysis, please explain what were the dependent variables and what 'family' you assumed for the distribution for each of these (e.g., for counts, Poisson / Negative Binomial).
  • In the description of the mixed effects analysis, please explain whether you used intercepts only or also used slopes (and why)
  • In the caption of figure 4 you talk about letters, where you probably mean digits.
  • On line 213 'extend' should be 'extent'
